# TNF-α Inhibitors Decrease Classical CD14^hi^CD16− Monocyte Subsets in Highly Active, Conventional Treatment Refractory Rheumatoid Arthritis and Ankylosing Spondylitis

**DOI:** 10.3390/ijms20020291

**Published:** 2019-01-12

**Authors:** Bogdan Batko, Agata Schramm-Luc, Dominik S. Skiba, Tomasz P. Mikolajczyk, Mateusz Siedlinski

**Affiliations:** 1Department of Rheumatology, J. Dietl Specialist Hospital, 31-121 Krakow, Poland; 2Department of Internal and Agricultural Medicine, Faculty of Medicine, Jagiellonian University Medical College, 31-121 Krakow, Poland; agata.schramm@uj.edu.pl (A.S.-L.); dominik.skiba@glasgow.ac.uk (D.S.S.); tomasz.mikolajczyk@glasgow.ac.uk (T.P.M.); mateusz.siedlinski@uj.edu.pl (M.S.); 3BHF Centre of Research Excellence, Institute of Cardiovascular and Medical Sciences, University of Glasgow, Glasgow G12 8TA, UK; 4Institute of Infection, Immunity and Inflammation, University of Glasgow, Glasgow G12 8TA, UK

**Keywords:** rheumatoid arthritis, ankylosing spondylitis, tumor necrosis factor inhibitor, disease activity, monocytes

## Abstract

Monocytes are pivotal cells in inflammatory joint diseases. We aimed to determine the effect of TNF-α inhibitors (TNFi) on peripheral blood monocyte subpopulations and their activation in ankylosing spondylitis (AS) and rheumatoid arthritis (RA) patients with high disease activity. To address this, we studied 50 (32 AS, 18 RA) patients with highly active disease with no prior history of TNFi use who were recruited and assigned to TNFi or placebo treatment for 12 weeks. Cytometric and clinical assessment was determined at baseline, four, and 12 weeks after initiation of TNFi treatment. We observed that treatment with TNFi led to a significant decrease in CD14^hi^CD16− monocytes in comparison to placebo, while circulating CD14^dim^CD16+ monocytes significantly increased. The TNFi-induced monocyte subset shifts were similar in RA and AS patients. While the percentage of CD14^dim^CD16+ monocytes increased, expression of CD11b and CD11c integrins on their surface was significantly reduced by TNFi. Additionally, CD45RA+ cells were more frequent. The shift towards nonclassical CD14^dim^CD16+ monocytes in peripheral blood due to TNFi treatment was seen in both AS and RA. This may reflect reduced recruitment of these cells to sites of inflammation due to lower inflammatory burden, which is associated with decreased disease activity.

## 1. Introduction

The significance of peripheral blood monocytes is increasingly recognized in the pathogenesis of inflammatory joint diseases, such as rheumatoid arthritis (RA) and ankylosing spondylitis (AS) [1,2,3]. Monocytes are divided into the predominant classical CD14^hi^CD16− population and CD16+ monocytes, which branch into intermediate CD14^hi^CD16+ and nonclassical CD14^dim^CD16+ subsets [4]. Treatment of arthritis aims to reduce tissue infiltrating monocytes, which are osteoclast precursors and sources of inflammatory cytokines [5]. CD16+ cells are increasingly present in both blood and synovium of RA patients, where they may express autoantigens correlating with the severity of joint damage [6]. Puchner et al. recently demonstrated in murine models that nonclassical monocytes are crucial in the development of arthritis, although literature also attributes a role for classical subsets in osteoclast differentiation and joint damage [7,8]. Monocyte-derived TNF-α may be a signaling molecule involved in this process and is a hallmark of inflammation [9]. When encountering immune complexes, nonclassical monocytes are selective producers of TNF-α and IL-1β [10] which have been investigated in numerous clinical trials as an effective method to reduce inflammatory disease activity [11,12]. CD16+ monocytes may fail to cross endothelial barriers at the onset of inflammation, while differentiating in peripheral blood and tissues due to perturbed cytokine production from classical CD14^hi^CD16− subsets [13]. Therefore, monocyte adhesion and activation markers are other emerging fields of interest. CD11b is involved in migration and adhesion. Polymorphisms of CD11b are associated with systemic lupus erythematosus and other immune complex mediated diseases [14]. Murine studies of targeted anti-CD11b antibodies demonstrate a suppressive effect on processes shaping arthritis [5]. To the best of our knowledge, peripheral blood monocyte heterogeneity has not been extensively studied in patients with RA or AS receiving anti-TNF-α treatment. Therefore, our goal was to evaluate the effect of anti-TNF-α treatment on peripheral blood monocyte subpopulations, their functional properties, and to compare the effect of treatment in high disease activity, and in conventional treatment-refractory AS and RA patients.

## 2. Results

Fifty patients were assigned to TNFi (*n* = 38) or placebo (*n* = 12) groups. Clinical characteristics of recruited patients did not show any baseline differences between groups according to factors presented in Table 1.

### 2.1. Effects of TNFi on Disease Activity

Disease activity at baseline and after 12 weeks of therapy was assessed. A significant decrease in disease activity was observed in the TNF-α inhibition arm compared to placebo. Respective parameters for RA and AS, namely DAS28-CRP and BASDAI, significantly decreased after 12 weeks of anti-TNF-α treatment but not in the placebo group. BASDAI index decreased after treatment by 3.57 points and increased by 0.36 points in the placebo group (*p* < 0.001). DAS28 index decreased by 2.07 points for TNFi and 0.61 for the placebo group (*p* < 0.001) (Appendix A).

### 2.2. Effects of TNFi on Monocyte Subpopulations

Monocyte subpopulations did not differ at baseline (t0) between the TNFi and placebo group (Figure 1B). We then investigated changes in the frequency of individual monocyte subpopulations during TNFi treatment, when compared to the placebo group. Patients treated with TNFi had no change in frequency of total monocytes in PBMC (delta t2−t0) as compared to placebo-treated patients. TNFi led to an increase of the nonclassical (CD14^dim^CD16+) monocyte fraction and a decrease of classical (CD14^hi^CD16−) subsets in comparison to placebo (Figure 2A,B). The same trend was observed for absolute monocyte numbers per microliter (data not shown). As this effect may be related to lower recruitment of inflammatory cells to the synovium, we studied the expression of monocyte markers involved in this process. Indeed, we observed reduced levels of CD11b and CD11c integrins on the surface of CD14^dim^CD16+ cells, which appeared to be most affected by TNFi treatment (Table 2 and Table 3). A higher fraction of CD45RA+ monocytes was observed in all monocyte subpopulations in the TNFi group (Table 3). Other monocyte subpopulations (CD14^hi^ and CD14^dim^CD16− cells) showed more discrete but consistent reductions of selected markers while having a tendency towards increased CD45RA and HLA-DR (Table 3).

### 2.3. Monocyte Responses to TNFi between AS and RA Patients

As the population studied here was heterogeneous and included both RA and AS subjects, we next performed a predefined analysis to observe differences in response to anti-TNF-α treatment between RA and AS patients (Figure 3). Analysis of monocytes and their subpopulations did not show significant differences in response to TNFi treatment between RA and AS patients. 

## 3. Discussion

The main finding of the present study is that in RA and AS patients, a decline in classical (CD14^hi^CD16−) monocytes was observed during anti-TNF-α treatment in comparison to placebo, while the amount of CD14^dim^CD16+ monocytes increased. This increase was associated with lower integrin expression on the surface of monocytes. A decline in the number of classical monocytes was previously reported in a small unrandomized study of 10 (5 RA, 5 AS) patients treated with infliximab, where a rapid decrease in classical monocyte subsets was reported [15]. Aeberli et al. suggested that TNFi treatment might restrict monocyte recruitment into inflamed tissues and TNFi-induced apoptosis in peripheral blood monocytes, which was also reported earlier [15,16]. Our observation that TNFi treatment increased circulating nonclassical (CD14^dim^CD16+) monocyte fractions to a greater extent than conventional treatment is also in line with the change seen in response to infliximab, although this phenomenon was unclear in AS patients (*p* = 0.045 at study endpoint) and mostly observed in RA patients [15]. This could lead to an interpretation that the TNFi effect on monocyte populations is closely determined by disease character. In our study, however, we provide evidence that the TNFi-mediated effect is present in both RA and AS. Moreover, we expand these observations to several TNF-α-inhibiting agents including etanercept, a soluble TNF-α receptor, and monoclonal antibodies other than infliximab. In response to TNF-α, murine models showed an increase in Ly6C^high^ monocytes, corresponding to human classical CD14^hi^CD16− monocytes, and a lesser increase in Ly6C^low^ subsets, similar to human nonclassical monocytes [17]. Subsequent treatment with adalimumab reduced circulating Ly6C^high^ but not Ly6C^low^, which is consistent with our findings.

Kawanaka et al. previously reported that elevated CD16 expressing monocyte subsets are characteristic for RA and the active disease state [18]. The authors suggested that overflow of cytokines from the inflamed joint promotes the differentiation of CD16+ monocytes. This subpopulation may further exacerbate the disease with increased chemokine receptor expression, subsequent infiltration, and pro-inflammatory cytokine production. When comparing RA patients to healthy controls, intermediate CD14^hi^CD16+ monocyte fractions are elevated, while nonclassical CD14^dim^CD16+ fractions are decreased [19]. In the present study, the TNFi-mediated shift towards nonclassical subsets and control of the intermediate fraction may resemble a change towards a “healthy” subset distribution. In AS, classical CD14^hi^CD16− monocyte fractions are increased and nonclassical CD14^dim^CD16+ are decreased when compared to healthy controls [20]. Similarly, the TNFi-induced rise in nonclassical subsets may result in regression from the monocyte “disease” distribution. The significance of this effect remains to be established.

While both diseases differ substantially in clinical features and blood monocyte heterogeneity, we demonstrated that in comparison to placebo, effective treatment with TNFi reduces disease activity and is paralleled by a shift in peripheral blood monocytes. In line with our findings, AS patients treated with anti-TNF-α agents showed a decline in circulating classical macrophages (M1) and a shift towards nonclassical macrophage (M2) subsets, corresponding with improvement in BASDAI index [21]. The M2/M1 shift positively correlated with joint destruction and negatively with inflammatory activity. This may explain why, even with effective control of disease activity by TNFi, erosive joint changes can still progress in some patients. This outcome is of high clinical importance. We hypothesize that the TNFi-induced shift toward nonclassical monocytes is sufficient to prevent elevation in the “inflammatory” intermediate phenotype and therefore reduces cytokine burden. However, the rise in the nonclassical subset, which has been demonstrated to be a mediator of tissue damage and correlates with joint destruction [8], can be seen as detrimental and counterintuitive. It should be noted that the “inflammatory” subpopulations of nonclassical and intermediate monocytes are observed to transmigrate less efficiently than classical subsets [13,22], but may lead to tissue and organ dysfunction [23]. We provide evidence that TNF-α-inhibiting agents may further reduce this capability by decreasing the expression of adhesive molecules, such as CD11b. Reduced monocyte tissue infiltration via TNFi-mediated effects on endothelium was previously proposed [15]. Enhanced CD11b expression occurs in response to chemotactic factors [24], while CD11b positive monocytes are able to differentiate into osteoclasts under TNF-α and Il-17 [25], at the same time as being inhibited by infliximab. Taken together, a significant decrease in CD11b expression in response to TNFi in all monocyte subpopulations, most prominent in nonclassical subsets (*p* < 0.001), indicates a reduced ability of these “disease-driving” cells to adhere to the endothelium and transmigrate. Indeed, antibodies targeting CD11b inhibit inflammatory cell recruitment, synovial infiltration, and suppress arthritis development in murine models [5]. The difficulty of nonclassical monocytes in crossing endothelial barriers may be amplified by TNFi-mediated effects, including CD11b downregulation. Finally, the increase in the “tissue-destructive” reservoir of nonclassical CD14^dim^CD16+ monocytes is sequestered in peripheral blood and limited from entering inflamed joints. This hypothesis is still preliminary and requires further study.

It was previously reported that in active AS (BASDAI ≥ 4), corresponding to our study subjects, greater CD11b expression was observed in patients with high disease activity compared to those with low disease activity [20]. Decreased expression of CD11b on monocyte subsets may stem from effective reduction of disease activity due to TNFi treatment, paralleled by a reduction in activation markers characteristic of active disease. CD11b is described as crucial in myeloid lineage differentiation, with TNF-α inhibition decreasing its expression on myeloid cells [26]. However, in RA, expression of CD11b is higher than in healthy controls and decreases after glucocorticoid treatment, which suggests that CD11b is an indicator of disease activity, rather than selectively tied to TNF-α inhibition [27]. We observed increased expression of CD45RA after TNFi treatment in our study. CD45RA, a monocyte activation marker, shows higher expression in healthy patients when compared to RA, particularly for intermediate subsets [19]. Response to adalimumab monotherapy has been associated with monocyte CD11c expression; future TNFi responders had higher basal CD11c expression than control and non-responders [28]. A shift towards the “healthy expression pattern” of monocyte markers was seen in clinical responders but was absent in non-responders. We observed a decrease in CD11c expression exclusively on CD14^dim^CD16+ cells after TNFi treatment. 

It has been suggested that the heterogeneity of inter-study findings may relate to diversity of treatment profiles (e.g., glucocorticoid use leading to decreased CD16+ expressing monocytes) [29]. Glucocorticoids were also shown to induce a phenotype change from classical CD14^hi^CD16− to CD16+ [30]. Whether MTX-naïve or MTX-treated patients differ with respect to monocyte distribution heterogeneity also remains unclear [31,32], though this may depend on whether the patient is MTX-responsive or MTX-refractory. To eliminate confounding factors in the present study, concomitant medication use, including glucocorticoids and MTX, did not differ between the TNFi and placebo groups and dosage was unaltered from baseline. 

The limitations of the study have to be addressed. Although the baseline monocyte subpopulations did not differ between the TNFi treatment and placebo group, a baseline discrepancy between activation markers was observed. Entirely homogenous populations were likely undifferentiated due to the small size of the placebo group (*n* = 12), limitations due to high disease activity, and other strict inclusion criteria. We approached this limitation by investigating three consecutive time points using repeated-measures ANOVA.

In summary, we observed a shift towards nonclassical CD14^dim^CD16+ monocytes in peripheral blood due to TNFi treatment, and this was seen in both AS and RA patients. To the best of our knowledge, this is the first placebo-controlled study investigating the effects of TNFi on monocyte subsets in AS and RA subjects. Our study contributes to a better understanding of the complex effects of TNFi in humans and could lead to the development of possible cellular biomarkers for TNFi monitoring. 

## 4. Materials and Methods 

### 4.1. Study Participants

Fifty patients with chronic inflammatory arthritis were recruited into the study; 32 were diagnosed with AS and 18 with RA. Diagnosis of AS was based on the modified New York criteria [33], while RA diagnosis was based upon ACR/EULAR 2010 criteria [34]. All patients had no prior history of anti-TNF-α treatment. At baseline, all patients had high disease activity. In AS patients this was defined as Bath Ankylosing Spondylitis Disease Activity Index (BASDAI) score of ≥4 in 2 assessments with a 12-week interval, while in RA patients this was defined as a 28-joint Disease Activity Score C-reactive protein (DAS28-CRP) of >5.1 measured twice with a 1-month interval. AS patients were recruited if they had an ineffective response to treatment with two nonsteroidal anti-inflammatory drugs (NSAIDs) consecutively for 3 months, at the maximum recommended or tolerated dose. RA patients treated ineffectively with 2 classical synthetic disease-modifying antirheumatic drugs (csDMARDs) for 6 months each, including the maximum recommended or tolerated doses of methotrexate for at least 3 months, were included. These criteria were in accordance with national reimbursement guidelines for biologic agents. Patients with a history of hepatitis, pneumonia, pyelonephritis within the last 3 months, opportunistic infection within the last 2 months, joint infection within the last year, neoplasm within the last 5 years, or pre-cancer stage were excluded from the study. Thirty-eight patients received anti-TNF-α treatment (24 AS, 14 RA), while 12 (8 AS, 4 RA) were given placebo. Our study originally aimed to recruit patients in a 3:1 design. Reimbursement availability for TNF-α inhibitor therapy at patient recruitment, following the fulfilment of inclusion criteria, was the determinant of patient allocation into either the active TNFi treatment arm or “waiting arm”, where they received placebo. Saline was used as placebo. We were unable to recruit a larger study group due to strictly defined eligibility, and limited resources. Clinical and laboratory investigators were blinded, and were not aware of the treatment received by patients. TNFi and placebo were administered by study nurses, who remained unblinded throughout the study, though they did not participate in any other aspects of the investigation. Patients received anti-TNF-α agents (Table 1) in the following dosage for 12 weeks: etanercept 50 mg s.c. weekly, adalimumab 40 mg s.c. every 2 weeks, certolizumab 400 mg s.c. every 2 weeks for a month, followed by 200 mg every 2 weeks, infliximab 3 mg/kg (RA) or 5 mg/kg (AS) i.v. 2 and 6 weeks from the first admission, then every 8 weeks. Baseline dosage of csDMARDs, glucocorticoids and NSAIDs were maintained throughout the study period. The study was conducted in accordance with the Declaration of Helsinki and Good Clinical Practice guidelines. The investigation protocol was approved by the Bioethics Committee of the Regional Chamber of Physicians in Krakow, Poland—investigation number 77/KBL/OIL/2013. Prior to enrolment, written consent was obtained from all patients. 

### 4.2. Blood Collection

Blood sample collection and clinical assessment were performed at baseline (*t* = 0), 4 (*t* = 1), and 12 (*t* = 2) weeks after initiation of anti-TNF-α treatment. Concentration of C-reactive protein (CRP) was determined using the immunoturbidimetric test.

### 4.3. Flow Cytometry

As described previously [19], peripheral blood mononuclear cells (PBMC) were isolated on the day of collection from peripheral EDTA-anticoagulated blood by density gradient centrifugation with LSM 1077 (Lymphocyte Separation Medium, PAA Laboratories GmbH, Cölbe, Austria). Cells (5.0 x 10^5^) were stained for 20 min on ice with fluorochrome-conjugated monoclonal antibodies (anti-CD14-APC-H7, anti-CD16-PE, anti-HLA-DR-PE-Cy7, anti-CD45RA-FITC, anti-CD11c-APC, and anti-CD11b-Pacific Blue; Becton Dickinson (BD) Biosciences—Pharmingen, San Diego, CA, USA) and then washed twice with phosphate buffered saline (PBS) containing 1% heat inactivated fetal bovine serum (FBS) (GIBCO). The cells were processed in the FACSCanto II flow cytometer (BD Biosciences, San Jose, CA, USA) and then analyzed with FlowJo software (TreeStar Inc., Ashland, OR, USA). Cells were gated in an SSC (side scatter)/FSC (forward scatter) plot with the scatter gate for monocytes partially extending into lymphocytes. The cells containing all monocytes and a part of the lymphocyte population were then gated in an HLA-DR/CD14 plot to exclude HLADR-negative natural killer cells (which would otherwise contaminate the CD14^dim^CD16+ subpopulation) and finally were analyzed for CD14 and CD16 expression [35]. Monocyte subsets were defined according to the expression of CD16 (Fcγ receptor type III) and CD14 (lipopolysaccharide receptor) as classical monocytes (CD14^hi^CD16−), intermediate monocytes (CD14^hi^CD16+), and nonclassical monocytes (CD14^dim^CD16+) [4] (Figure 1A). The expression of β2-integrins (CD11b and CD11c), CD45RA, and HLA-DR on monocyte subsets was quantified. Results are presented as percentages and mean fluorescence intensity (MFI). 

### 4.4. Statistics

Analyses concerning over-time changes in cell characteristics were performed using repeated measures ANOVA, where interaction between time and patient group (anti-TNF-α treatment vs. placebo) was tested. Differences between groups at baseline were tested using the Student’s *t*-test (continuous variables) or chi-squared test (categorical variables). Analyses were performed using IBM SPSS Statistics software (ver. 25). Differences were considered statistically significant for *p* < 0.05.

## Figures and Tables

**Figure 1 ijms-20-00291-f001:**
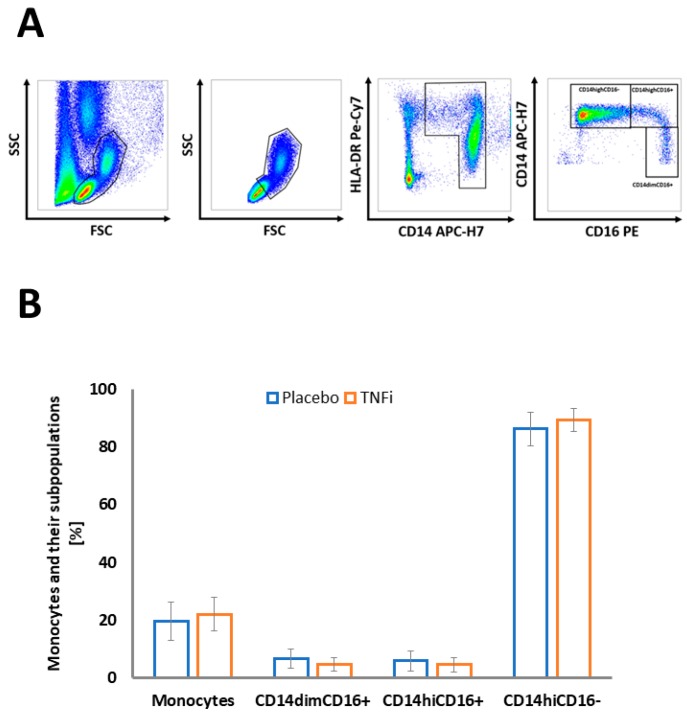
(**A**) Gating strategy of monocyte subpopulation in circulating blood. (**B**) Mean monocyte content and their subpopulation distribution at baseline (t0) in placebo and the anti-TNF-α treated group (TNFi). Error bars represent SD.

**Figure 2 ijms-20-00291-f002:**
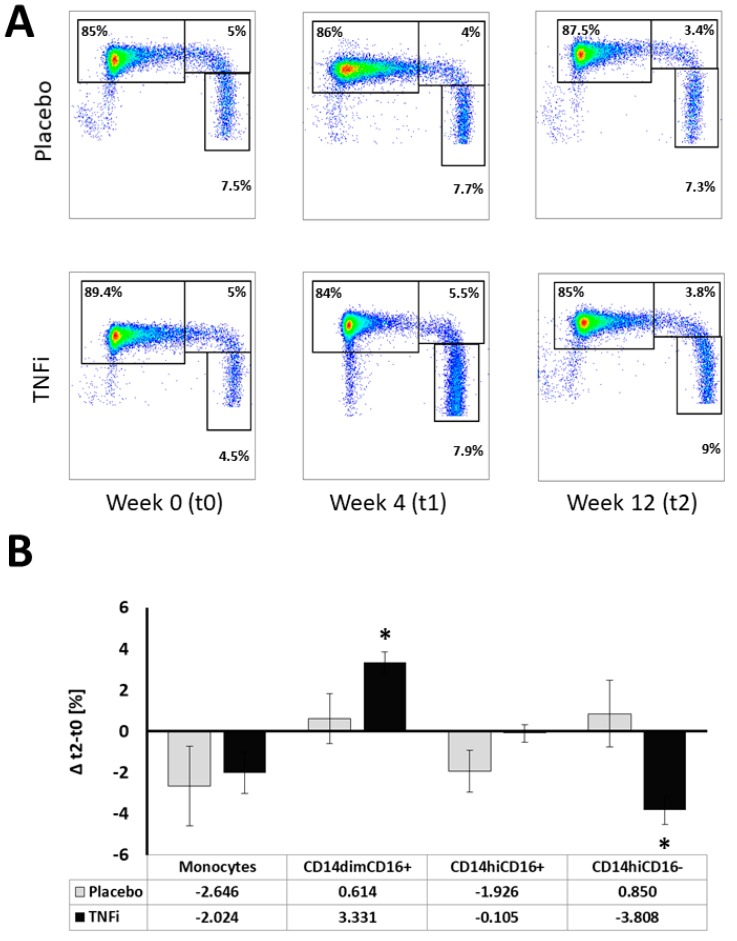
Monocytes and their subpopulation content in placebo and TNFi groups. (**A**) Representative flow cytometric dot plots for placebo and TNFi treated groups at week 0 (t0), 4 (t1) and 12 (t2). Panel (**B**): Mean monocyte content and their subpopulation distribution was calculated as a delta t2−t0. Error bars represent SEM; * *p* < 0.05.

**Figure 3 ijms-20-00291-f003:**
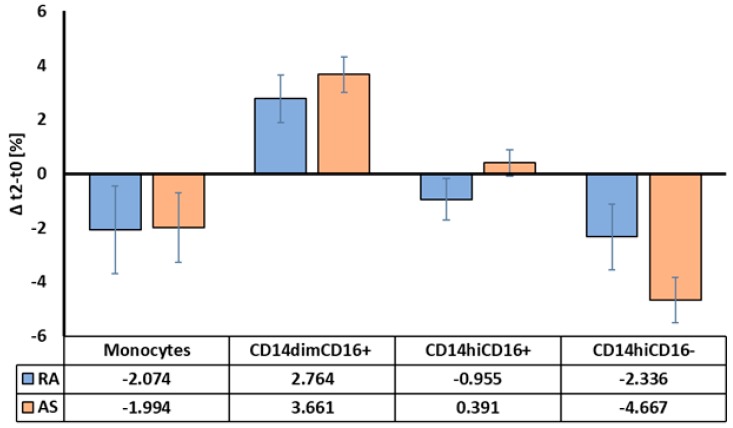
Monocytes and their subpopulations in RA and AS groups treated with TNFi. Data was calculated as a delta t2−t0. Error bars represent SEM.

**Table 1 ijms-20-00291-t001:** Baseline clinical and biochemical characteristics of rheumatoid arthritis (RA) and ankylosing spondylitis (AS) patients.

Clinical Characteristic	Anti-TNF-α Treatment (*n* = 38) Mean (SD)	Placebo Comparator (*n* = 12) Mean (SD)	*p* Value
**Demographic Factors**			
Age (years)	40.37 (11.74)	42.75 (7.40)	0.41
Gender (%)			
Male (n)	57.89% (22)	58.33% (7)	0.98
Female (n)	42.11% (16)	41.67% (5)	0.98
Smoking history			
Never smoker (n)	42.11% (16)	33.33% (4)	0.85
Ever smoker (n)	34.21% (13)	41.67% (5)	0.85
Active smoker (n)	23.68% (9)	25.00% (3)	0.85
BMI (kg/m2)	27.04 (4.05)	26.89 (4.43)	0.92
**Disease Related Factors**			
Duration of the disease (years)	8.37 (7.56)	13.46 (10.50)	0.14
CRP (mg/l)	21.16 (38.67)	12.46 (9.55)	0.21
BASDAI (AS), *n* = 32	7.14 (1.05)	6.34 (1.75)	0.25
HLA-B27 (AS)	87.5% (*n* = 21)	100% (*n* = 8)	0.29
DAS 28 (RA), *n* =18	5.50 (0.71)	5.21 (0.22)	0.13
RF (RA)	71.43% (*n* = 10)	100% (*n* = 4)	0.23
aCCP (RA)	78.57% (*n* = 11)	75% (*n* = 3)	0.31
**Drugs**			
csDMARDS			
Methotrexate (n)	39.47% (15)	25% (3)	0.36
Sulfasalazine (n)	18.42% (7)	25% (3)	0.62
Leflunomide (n)	2.6% (1)	0% (0)	0.57
Glucocorticosteroids	39.47% (15)	41.67% (5)	0.89
NSAIDs	71.05% (27)	83.33% (10)	0.40
Anti-TNF-α			
Etanercept (n)	50.00% (19)	-	
Adalimumab (n)	21.05% (8)	-	
Infliximab (n)	21.05% (8)	-	
Certolizumab (n)	7.89% (3)	-	

Data are shown as means ± SD or percentages (*n*). RA: rheumatoid arthritis; AS: ankylosing spondylitis; DAS28-CRP: disease activity score in 28 joints using CRP; BASDAI: Bath Ankylosing Spondylitis Disease Activity Index; csDMARDSs: classical synthetic Disease-modifying antirheumatic drugs; CRP: C-reactive protein. For numeric data, Student’s *t*-test was performed; for categorical data, chi-squared test was used.

**Table 2 ijms-20-00291-t002:** Subpopulations of monocytes and their activation markers in different time t0, t1 and t2 in placebo and anti-TNF-α treatment.

Subpopulation	Placebo			Anti-TNF-α Treatment			*p* Value
Timepoint	t0	t1	t2	t0	t1	t2	
Monocytes (%)	19.44 (6.69)	18.97 (5.70)	16.80 (5.69)	21.98 (5.81)	18.15 (6.23)	19.96 (5.96)	*p* = 0.124
CD45RA MFI	3188.08 (1223.47)	3040.25 (914.15)	3033.00 (822.31)	2641.50 (1580.84)	3484.53 (1806.13)	3654.32 (2189.89)	***p* = 0.013**
CD11c MFI	778.92 (140.42)	901.08 (234.06)	877.83 (229.42)	937.24 (407.19)	972.34 (334.35)	980.74 (399.46)	*p* = 0.832
CD11b MFI	900.58 (212.85)	873.17 (166.56)	834.33 (227.42)	3310.42 (2993.18)	4138.79 (4056.15)	2100.51 (2861.03)	*p* = 0.121
HLA-DR MFI	10984.50 (3181.69)	12183.08 (4130.38)	11438.00 (4321.06)	9481.24 (4661.29)	12162.82 (6078.35)	13064.74 (5250.11)	*p* = 0.274
CD14^dim^CD16+ (%)	6.52 (3.41)	7.28 (4.29)	7.14 (4.13)	4.65 (2.34)	7.43 (4.53)	7.98 (3.57)	*p* = 0.073
CD45RA MFI	11065.17 (5181.33)	10355.00 (5370.98)	10033.42 (4897.20)	9254.92 (6000.12)	10994.37 (5624.30)	11494.16 (6599.58)	***p* = 0.049**
CD11c MFI	2173.75 (318.64)	2665.25 (573.03)	2813.00 (868.16)	2998.82 (1273.40)	2756.58 (1124.21)	2690.21 (1206.80)	***p* = 0.046**
CD11b MFI	531.92 (106.24)	556.17 (115.43)	535.50 (123.92)	1943.86 (1665.57)	1931.00 (1835.86)	1012.83 (1084.80)	***p* = 0.028**
HLA-DR MFI	25946.50 (4698.43)	29249.33 (7221.83)	27596.17 (6016.55)	24027.34 (10224.44)	24758.55 (16111.57)	26336.08 (17167.62)	*p* = 0.771
CD14^hi^CD16+	5.71 (3.54)	5.03 (2.86)	3.79 (2.12)	4.41 (2.40)	4.97 (3.51)	4.31 (1.97)	*p* = 0.170
CD45RA MFI	2871.33 (1139.52)	3070.92 (1566.17)	2804 (1197.48)	3046.66 (2779.49)	3371 (2334.81)	3456.95 (3063.03)	*p* = 0.593
CD11c MFI	1838.08 (394.12)	2232.67 (586.69)	2193.83 (478.66)	2458.05 (999.10)	2417 (796.66)	2435.16 (748.38)	*p* = 0.315
CD11b MFI	1084.00 (266.68)	1068.25 (209.36)	1045.50 (265.60)	4071.86 (3416.93)	5051.47 (4754.07)	2548.31 (3371.88)	*p* = 0.102
HLA-DR MFI	49088.17 (11819.53)	54980.00 (11033.93)	54020.42 (15785.05)	45712.00 (19544.37)	49330.89 (19217.99)	55932.68 (21573.64)	*p* = 0.466
CD14^hi^CD16− (%)	86.28 (5.77)	86.17 (6.87)	87.13 (5.93)	89.35 (4.04)	85.19 (7.47)	85.54 (4.50)	***p* = 0.026**
CD45RA MFI	2002.58 (671.09)	1935.42 (510.77)	1853.42 (515.27)	1862.82 (1281.84)	1831.71 (968.78)	1997.26 (1569.28)	*p* = 0.297
CD11c MFI	588.833 (122.13)	663.25 (144.66)	644.667 (126.93)	738.789 (340.83)	719.447 (254.83)	723.763 (307.49)	*p* = 0.725
CD11b MFI	909.75 (220.24)	883.25 (173.51)	841.42 (231.86)	3355.67 (3058.44)	4317.13 (4268.82)	2210.06 (3167.63)	*p* = 0.139
HLA-DR MFI	7218.33 (2526.62)	8054.08 (2533.40)	7921.17 (2929.59)	6706.42 (3353.68)	8473.97 (3954.01)	9368.45 (3641.07)	*p* = 0.340

Data are shown as mean and standard deviation (SD). Statistical analysis was performed using ANOVA of repeated measures. *p* Value shown for interaction term (time x group). *p* Value lower than 0.05 marked as a bold.

**Table 3 ijms-20-00291-t003:** Expression of activation markers on monocytes and their subpopulation shown as a delta t2−t0.

	Placebo	TNFi	*p* Value
Monocytes			
CD11b (%)	−1.08 (2.10)	−3.51 (3.52)	**0.008**
CD11b, MFI	−66.25 (379.01)	−1540.21 (2789.60)	**0.006**
CD11c (%)	1.69 (8.58)	−0.84 (9.88)	0.400
CD11c, MFI	98.92 (289.18)	43.50 (482.08)	0.631
CD45RA (%)	−3.48 (7.86)	3.58 (10.44)	**0.020**
CD45RA, MFI	−155.08 (689.07)	1012.82 (1203.48)	**<0.001**
HLA-DR (%)	−0.05 (0.26)	0.55 (1.73)	**0.045**
HLA-DR, MFI	453.50 (3829.02)	3583.50 (6990.94)	0.056
CD14^dim^CD16+			
CD11b (%)	−5.83 (15.24)	−21.97 (27.99)	**0.019**
CD11b, MFI	3.58 (120.93)	−1071.09 (1252.97)	**<0.001**
CD11c (%)	0.08 (0.20)	0.08 (0.26)	0.923
CD11c, MFI	639.25 (868.90)	−308.61 (1395.26)	**0.009**
CD45RA (%)	−0.10 (0.47)	1.58 (4.49)	**0.030**
CD45RA, MFI	−1031.75 (3329.37)	2239.24 (4509.90)	**0.012**
HLA-DR (%)	−0.08 (0.22)	−0.06 (1.35)	0.950
HLA-DR, MFI	1649.67 (4163.04)	2308.74 (18560.71)	0.840
CD14^hi^CD16+			
CD11b (%)	−0.44 (0.72)	−0.63 (1.12)	0.513
CD11b, MFI	−38.50 (424.72)	−1841.12 (3118.36)	**0.003**
CD11c (%)	−0.29 (1.03)	0.05 (1.46)	0.378
CD11c, MFI	355.75 (582.27)	−22.90 (1065.56)	0.125
CD45RA (%)	−2.72 (4.21)	3.50 (8.57)	**0.002**
CD45RA, MFI	−67.33 (884.13)	410.29 (1210.45)	0.150
HLA-DR (%)	−0.80 (1.20)	−2.00 (6.96)	0.314
HLA-DR, MFI	4932.25 (9930.15)	10220.68 (25135.67)	0.294
CD14^hi^CD16−			
CD11b (%)	−0.17 (0.22)	−0.52 (0.57)	**0.004**
CD11b, MFI	−68.33 (392.09)	−1504.85 (3016.97)	**0.011**
CD11c (%)	2.19 (9.88)	−1.14 (11.63)	0.341
CD11c, MFI	55.83 (197.20)	−15.03 (409.84)	0.423
CD45RA (%)	−4.20 (9.28)	2.37 (11.31)	0.055
CD45RA, MFI	−149.17 (234.61)	134.45 (627.74)	**0.025**
HLA-DR (%)	0.00 (0.27)	0.67 (1.74)	**0.026**
HLA-DR, MFI	702.83 (2131.76)	2662.03 (4829.73)	0.056

Data are shown as mean and standard deviation (SD). *p* Value lower than 0.05 marked as a bold.

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
