# Peer review of "TNF-α Inhibitors Decrease Classical CD14hiCD16− Monocyte Subsets in Highly Active, Conventional Treatment Refractory Rheumatoid Arthritis and Ankylosing Spondylitis"

_ijms, 2019, doi:10.3390/ijms20020291_

Reviewer 1 Report

The subject of the article is very interesting, once the effect of anti-TNF-α 58 treatment  was analyzed. However, I did not see reference in the approval of the ethics committee, without this approval the article cannot be accepted?

General comments:

1-     Why Fifty patients, TNFi (n=38) and placebo (n=12) ?

2-     In the figure S1 does not understand the statistics, there are differences between t0 e t2? modify the line

3-     In figure 1,2,3 you can fill the bars with different patterns.

4-     In the Figure 2 and 3 the Error bars represent are too large, could use the standard error and not the SD or increase the number of patients.

5-     Standardize tables, statistical differences or are in bold or underlined.

Author Response

We would like to thank the Reviewer 1 for time and valuable input in reviewing our manuscript.

Point 1: …I did not see reference in the approval of the ethics committee, without this approval the article cannot be accepted?

Response 1: We provided information regarding ethics committee approval in lines 331-334 (333-336 in revised version): “The investigation protocol was approved by the Bioethics Committee of the Regional Chamber of Physicians in Krakow, Poland – investigation number 77/KBL/OIL/2013. Prior to enrolment, written consent was obtained from all patients.”

Point 2: Why Fifty patients, TNFi (n=38) and placebo (n=12) ?

Response 2: The limited placebo group was related to the allocation of patients into treatment/control groups by availability of TNF inhibitors. TNF inhibitor treatment in Poland is limited by strict and extensive reimbursement criteria defining disease activity and
inadequate conventional treatment response. Secondly, the number of patients for which treatment will be accessible and reimbursed is not easily predictable (“constant”). Patients eligible for TNF inhibitor therapy in Poland experience a particularly active and refractory course of disease, for which prompt initiation of treatment is a priority. Methodologically, during recruitment we hoped to achieve a 3:1 design of allocation, however, treatment was accessible for more patients, for whom it would have been unethical to maintain conventional therapy. It should be noted that we were expecting to recruit a larger study group, however, our resources were limited.

The original excerpt reads as follows, “Due to limited reimbursement, patients who met the inclusion criteria were assigned according to treatment availability at recruitment, into either the active treatment arm or “waiting arm,” where they received placebo. Saline was used as placebo.”

We revised this fragment to,
“Our study originally aimed to recruit patients in a 3:1 design. Reimbursement availability for TNF inhibitor therapy at patient recruitment, following the fulfilment of inclusion criteria, was the determinant of patient allocation into either the active TNFi treatment arm or “waiting arm”, where they received placebo. Saline was used as placebo. We were unable to recruit a larger study group due to strictly defined eligibility, and limited resources.”

Point 3: In the figure S1 does not understand the statistics, there are differences between t0 e t2? modify the line

Response 3: In figure S1, we observed differences between t2 and t0 for the TNFi group in both panel A and panel B, indicating a decrease of disease activity following TNFi treatment, but not Placebo. In the line 104 there is a misspelling “t12’ instead of t2. Correction was made in original version.

Point 4: In figure 1,2,3 you can fill the bars with different patterns.

Response 4: Thank you for this insight. We changed the colour scheme of figure 2B for grey and black bars to differ from Figure 1 and Figure 3.

Point 5: In the Figure 2 and 3 the Error bars represent are too large, could use the standard error and not the SD or increase the number of patients.

Response 5: Again, thank you. We made changes to the error bars to show SEM instead of SD in Figure 2 and 3. Labelling has been updated.

Point 6: Standardize tables, statistical differences or are in bold or underlined.

Response 6: Corrections were made to the original version and statistical differences are now labelled in bold.

Reviewer 2 Report

In this study, the authors investigated the effect of TNFi in the treatment of AS and RA. The authors used the cytometric and clinical assessment of the TNFI at baseline, 4, and 12 weeks.

The study was well performed and the findings should be of interest to the journal's readership.

During treatment with TNFi led to a significant decrease in CD14hiCD16- monocytes in comparison to placebo, while circulating CD14 25 dimCD16+ monocytes significantly increased, but the percentage of CD14 27 dimCD16+ monocytes increased, expression of CD11b and

  CD11c integrins on their surface was significantly reduced. The authors have shown changes in  CD14 29 dimCD16 + monocytes in peripheral blood due to TNFs was seen in both AS and RA. The results obtained may be of interest to rheumatologists.

Author Response

We would like to thank the Reviewer 2 for time and valuable input in reviewing our manuscript.

Point 1: In this study, the authors investigated the effect of TNFi in the treatment of AS and RA. The authors used the cytometric and clinical assessment of the TNFI at baseline, 4, and 12 weeks. The study was well performed and the findings should be of interest to the journal's readership. During treatment with TNFi led to a significant decrease in CD14hiCD16- monocytes in comparison to placebo, while circulating CD14dimCD16+ monocytes significantly increased, but the percentage of CD14dimCD16+ monocytes increased, expression of CD11b and CD11c integrins on their surface was significantly reduced. The authors have shown changes in CD14dimCD16 + monocytes in peripheral blood due to TNFs was seen in both AS and RA. The results obtained may be of interest to rheumatologists.

Response 1: Thank you for your valuable comment. We appreciate your interest in our work

Round  2

Reviewer 1 Report

The authors have revised the manuscript adequately according to my comments and suggestions.